# The Impact of Mitochondrial Deficiencies in Neuromuscular Diseases

**DOI:** 10.3390/antiox9100964

**Published:** 2020-10-09

**Authors:** Judith Cantó-Santos, Josep M. Grau-Junyent, Glòria Garrabou

**Affiliations:** 1Muscle Research and Mitochondrial Function Laboratory, CELLEX-IDIBAPS, Faculty of Medicine, University of Barcelona, 08036 Barcelona, Spain; jcanto@clinic.cat (J.C.-S.); jmgrau@clinic.cat (J.M.G.-J.); 2Internal Medicine Department, Hospital Clinic of Barcelona, 08036 Barcelona, Spain; 3CIBERER—Spanish Biomedical Research Centre in Rare Diseases, 28029 Madrid, Spain

**Keywords:** neuromuscular diseases (NMDs), mitophagy, reactive oxygen species (ROS), damage-associated molecular patterns (DAMPs), oxidative stress, mitochondrial respiratory chain (MRC)

## Abstract

Neuromuscular diseases (NMDs) are a heterogeneous group of acquired or inherited rare disorders caused by injury or dysfunction of the anterior horn cells of the spinal cord (lower motor neurons), peripheral nerves, neuromuscular junctions, or skeletal muscles leading to muscle weakness and waste. Unfortunately, most of them entail serious or even fatal consequences. The prevalence rates among NMDs range between 1 and 10 per 100,000 population, but their rarity and diversity pose difficulties for healthcare and research. Some molecular hallmarks are being explored to elucidate the mechanisms triggering disease, to set the path for further advances. In fact, in the present review we outline the metabolic alterations of NMDs, mainly focusing on the role of mitochondria. The aim of the review is to discuss the mechanisms underlying energy production, oxidative stress generation, cell signaling, autophagy, and inflammation triggered or conditioned by the mitochondria. Briefly, increased levels of inflammation have been linked to reactive oxygen species (ROS) accumulation, which is key in mitochondrial genomic instability and mitochondrial respiratory chain (MRC) dysfunction. ROS burst, impaired autophagy, and increased inflammation are observed in many NMDs. Increasing knowledge of the etiology of NMDs will help to develop better diagnosis and treatments, eventually reducing the health and economic burden of NMDs for patients and healthcare systems.

## 1. Introduction

### 1.1. Neuromuscular Diseases (NMDs)

NMDs are a heterogeneous group of acquired or inherited rare disorders caused by injury or dysfunction of the anterior horn cells of the spinal cord (lower motor neurons), peripheral nerves, neuromuscular junctions (NMJ), or skeletal muscles, resulting in muscle weakness and waste, swallowing and breathing difficulties, and cardiac failure [1]. NMDs share some common features: weakness, twitching, torpidity, and cramps but signs and symptoms of various NMDs might be very different [2]. Unfortunately, most of them entail serious or even fatal consequences. 

NMDs exhibit a diversity of symptoms due to large genetic and clinical heterogeneity, as more than 500 genes are implicated in their pathogenesis, setting difficulties for healthcare and research [2,3]. Most NMDs’ prevalence ranges between 1 and 10 per 100,000 population [4]. Given the rarity and diversity of NMDs, patients experience long delays in diagnosis (typically 7 years) and 30% are never achieved [1], probably because many of them lack non-invasive diagnostic and prognostic biomarkers. Likewise, models of disease or therapeutic options are infrequent, but growing interests and knowledge are focused in depicting the etiology of these disorders [5].

Currently, diagnosing NMD starts with clinical observation, which can identify atrophy or loss of muscle bulk or tone. Blood tests can determine abnormal levels of various common metabolites and antigens in certain NMD. In addition, electromyography (EMG) helps to assess the health of muscles and lower motor neuron nerve cells that control those muscles to diagnose a number of NMD [6]. Once muscle affection is assured, multitude of diagnostic algorithms are developed to establish the specific disorder. Usually, muscle biopsy takes part in the diagnostic clues. Depending on the clinical and pathological presentation, the study of mitochondrial DNA (mtDNA) genome and mitochondrial respiratory chain (MRC) function is also required [7]. This fact indicates the mitochondrial etiology of some NMDs and why mitochondrial deregulation may play a role in the rest. 

Despite the challenges in NMD diagnosis, genetic diagnosis of these diseases has improved over the last 20 years [8]. Advances in sequencing and understanding the human genome has helped to detect monogenic NMD. Improved diagnosis and increased life expectancy may rise the prevalence of NMD worldwide while massive genome sequencing helps to depict complex disease inheritance and etiology. In addition, research in pathways affected in NMD is essential to improve the treatments and guide toward more personalized therapies in NMDs.

#### Classification of NMDs

The knowledge about new NMDs is continually increasing and changing. Currently, there are 7 main groups of NMDs, represented in Figure 1, which are [6,9,10]:Muscular dystrophies (MD): affect the structure of the muscle cells, causing weakness and degeneration of the skeletal muscles. MD subtypes are described in Table 1.Myopathies other than dystrophies: affect tone and contraction of muscles controlling voluntary movements; may include inflammation of muscles or related tissues, resulting in muscular weakness. Numerous myopathies have been described and classified in different groups (Table 2).Neuromuscular junction (NMJ) diseases: result from the destruction, dysfunction, or absence of one or more key proteins involved in the transmission of signals between nerves and muscles (Table 3).Motor neuron diseases: involve nerve cells in the spinal cord (lower motor neurons). Lower motor neurons progressively lose their function, causing the muscles they control to become weak and eventually non-functional (Table 3).Peripheral nerve diseases: involve motor and sensory nerves that connect the brain and spinal cord to the rest of the body causing impaired sensations, movement or other functions (Table 4).Mitochondrial diseases: involve errors in metabolism that affect energy production in muscle cells (Table 4).Ion channel diseases: diseases associated with defects in proteins forming ion channels, leading to muscular weakness, absent muscle tone, or episodic muscle paralysis (Table 5).

### 1.2. Muscle and Nerve System

As outlined in the classification of NMDs, either muscle or nerve dysfunction can result in a NMD. Both tissues are formed by highly specialized cells; neurons in the peripheral nervous system and muscle cells (organized into sarcomeres) in skeletal muscles. Although nerve and muscle tissues contain a reservoir of stem cells for tissue remodeling and regeneration (called satellite cells in muscles), the rate of cell and tissue renewal is quite low, and neurons and sarcomeres, once formed, never proliferate. This is one of the features that causes nerve and muscle tissue disease in case of cell injury.

One of the reasons for cell injury is the strong dependence that nerve and muscle tissue have on mitochondrial function. Their high energetic needs are almost entirely sustained by oxidative metabolism (high-energy demands provided by mitochondrial oxidative respiration). Thus, any deficiency at mitochondrial level may endanger cell and tissue function, being the base of NMDs.

At structural level, the interaction of nervous and musculoskeletal system occurs in the NMJ, also called motor end plate [54]. Muscles are organized in bundles, called muscle fibers, made up of myocytes (muscle cells). Muscle fibers together with lower motor neurons form the motor unit. To reach motor units, nerve impulses travel from the central nervous system (CNS) to the spinal cord and to the whole body within peripheral nerves [86]. In Figure 1, a motor unit is represented together with the classification of the main NMDs.

Muscle fibers are innervated with motor neuron axon terminals. The communication of muscle cells with lower motor neurons is mainly based in changes in the membrane potential and ion channel opening and closing. Briefly, when lower motor neurons are stimulated, their membrane potential changes due to exchange of sodium (Na^+^) and potassium (K^+^) across the cell membrane, prompting the depolarization of the neuron. Inside the neuron, Na^+^ entrance through specific channels promotes the release of calcium (Ca^2+^) ions from the endoplasmic reticulum (ER) into the cytosol, triggering the further release of neurotransmitters out of nerve cells. These neurotransmitters are sensed by muscle cells, promoting the contraction of muscle fibers [87]. Muscle contraction is mainly controlled by Ca^2+^ fluxes.

Ca^2+^ increase in the cytosol of muscle cells allows actin and myosin (contractile proteins), to interact, causing muscle contraction. Right after, Ca^2+^ are pumped back into the ER causing muscle relaxation [87]. Overall, body movement is actioned by muscle fibers controlled by the nervous system.

Energetic supply is essential for ion exchange and NMJ functioning. Based on energetic criteria and mitochondrial metabolism, muscle fibers can be divided into two categories: type I (slow-twitch) fibers and type II (fast-twitch) fibers. Type I fibers contain more mitochondria and myoglobin than type II fibers. Mitochondria generate ATP by cellular respiration, for which oxygen is required and provided by myoglobin, which stores and carries oxygen in muscle cells. Type II fibers synthesize the majority of their energy through anaerobic respiration, without the need of oxygen, a faster process than mitochondrial respiration but that provides lower amount of ATP [88].

Overall, the CNS and muscle require high amount of energy, mostly produced in mitochondria, and that is why mitochondrial defects are causal or remarkable in the development of NMDs [14]. As mitochondria are essential in nerve and muscle tissue function, primary mitochondrial diseases, caused by mutations in a mitochondrial gene, generally manifest as encephalomyopathies [89], mainly affecting brain and muscle tissues. This is the case of mitochondrial encephalomyopathy with lactic acidosis syndrome (MELAS) or myoclonic epilepsy with ragged-red fibers (MERRF). NMD such as Charcot-Marie-Tooth neuropathy type 2A (CMT2A) [74] or amyotrophic lateral sclerosis (ALS) can also be caused by mitochondrial mutations (CMT2A can be caused by mutations in MFN2 and ALS by mutations in SOD1) [59]. Interestingly, mitochondria can also play a secondary role in the development of the rest of NMD when the mutation or deficiency is not directly related or located in the mitochondria, since affected cells will need additional ATP to support homeostatic mechanisms disbalanced (anti-stress or antioxidant responses), while minimizing the production of ROS. If mitochondria are unable to counterbalance cell dysfunction, a secondary mitochondrial disease will appear, like spinal muscular atrophy (SMA) (usually caused by a mutation in SMN1) [71] or Duchenne muscular dystrophy (DMD) due to mutation in DMD) [90]. Therefore, mitochondrial function is key in the onset or progression of most of NMDs, regardless of their genetic origin.

### 1.3. Mitochondrial Function

Despite their clinical and pathological diversity, some overlaps can arise among all NMDs. One of them are molecular events revealing metabolic alterations related to the mitochondria. Although originally considered the “powerhouse of the cell” [91], mitochondria are organelles with a central role in multiple cellular processes: ATP production, apoptosis, β-oxidation of fatty acids, iron-sulfur cluster synthesis and in a broader range of signaling pathways. Additionally, in case of troubleshooting, mitochondria are the main center of ROS production and cell death through the control of apoptosis [92]. Thus, mitochondrial health is essential for cell function and any misbalance in bioenergetics can endanger cell, tissue, and patient survival [93].

Mitochondria can multiply their number in case of energy need or survival of cells, through a process called mitochondrial biogenesis [94]. In addition, mitochondria are dynamic organelles that constantly fuse and divide to increase their energy supply (fusion) or target the mitochondria for degradation (fission). This process is called mitochondrial dynamics and triggers changes in mitochondrial number, morphology, and size [92]. 

Even if the number of mitochondria rises in a cell, impairment of MRC increases the production of ROS [95], which increases the oxidative stress levels in cells, activating mechanisms to counterbalance this excessive ROS. These repair mechanisms start with autophagy and mitophagy. Autophagy is the clearance of damaged organelles and protein aggregates by lysosomal degradation; while mitophagy is a specialized autophagy that degrades dysfunctional mitochondria to maintain mitochondrial integrity [96]. Both processes remove the excess of ROS produced in mitochondria. Excessive ROS can also affect the ER, as it is one of the main sensors of cellular stress. ER and mitochondria are connected through mitochondria-ER membrane contact sites, called mitochondria-associated membranes (MAMs). When ER senses ROS, ER redox status can change, and might induce ER stress [97]. To defend against ER stress and overcome potential protein misfolding, cells can activate an adaptive unfolded protein response (UPR), which increases the size and capacity of ER to reduce ER stress and increase the removal of misfolded proteins by autophagy [98].

In addition, excessive ROS can trigger an intracellular inflammatory response [99] which has an impact in mitochondrial genomic instability and respiratory chain dysfunction [100]. ROS burst, impaired autophagy, and increased inflammation are observed in many NMDs [101], whose prevalence increases with aging. However, these mechanisms have not been thoroughly revised in all groups of NMD and have not been compared before.

Overall, the aim of this review is to point out the role of excessive oxidative stress in mitochondrial function and its impact in autophagy and inflammation in NMDs (Table 1, Table 2, Table 3, Table 4 and Table 5). Increasing knowledge of the etiology of NMDs will help to develop better diagnosis and treatments to modify the natural history of these chronic disabling diseases, eventually reducing both health and economic burdens of NMDs for patients and healthcare systems [9]. 

## 2. Mitochondrial Pathways Altered in NMD

### 2.1. Mitochondrial Genome and mtDNA Mutations

Mitochondria contain their own genome, which encodes 13 proteins, 22 tRNAs, and 2 rRNAs. The remaining proteins involved in mitochondrial structure and function are encoded by the nuclear DNA, translated in the cytosol and translocated into mitochondria [92]. Although mtDNA only encodes ±1% of the mitochondrial proteins, these elements are critical components of the MRC, essential for mitochondrial function [94]. Nuclear and mitochondrial intergenomic communication is essential to build the MRC complexes that enables cell bioenergetics and promotes healthy organ function [102]. 

If mtDNA mutates, these mutations can be present in all mtDNA copies in the cell (homoplasmy) or only in a fraction (heteroplasmy). Almost all humans carry low levels of heteroplasmic mtDNA point mutations, which could be pathogenic [103]. Both the subcellular distribution of mutated mtDNA and the intrinsic pathogenicity of the mutation itself may also be important in determining whether a mtDNA defect is expressed phenotypically [7]. mtDNA mutations are also related to the aging process or age-associated diseases [94].

The distribution of a mutation in different tissues may change with time depending on the mitotic activity of the tissue [92]. In tissues with ongoing cell division, there is a chance that a defect in mtDNA may be selected in or out. This may explain the predilection for mitochondrial diseases to manifest clinically in postmitotic tissues such as the central nervous system and skeletal muscle. This mechanism may also explain why some mitochondrial phenotypes change with time [7,104].

Pathogenic mtDNA mutations must be present in 60–90% of mitochondria in a cell to trigger MRC dysfunction [105]. Computational models estimate that only mtDNA mutations generated in early life could expand to reach the threshold and cause MRC dysfunction in aging human adults [106]. One of the reasons is because mtDNA is organized into mitochondrial nucleoids [107] and coupled with mitochondrial transcription factor A (TFAM), to make mtDNA less accessible to mutations than naked DNA. In fact, lower amount of TFAM is found in hypothyroid myopathy (endocrine myopathy, Table 2), triggering reduced mtDNA copy number and mitochondrial dysfunction [38]. 

Other NMDs associated to mtDNA mutations are metabolic myopathies, ALS, CMT2, mitochondrial myopathies, and Friedreich’s ataxia (Table 2, Table 3 and Table 4).

Apart from mtDNA point mutations, single large deletion in mtDNA at high levels cause multisystem diseases in children and NMDs associated with chronic progressive external ophthalmoplegia in adults and proximal myopathy [76,79]. Other NMDs with mtDNA deletions are sporadic inclusion body myositis (sIBM) (reported in 67% of sIBM patients) [108,109] and giant axonal neuropathy (GAN). These deletions are generally spontaneously acquired [105], probably as accidents during mtDNA replication in the oocyte, in the early embryo or later on, along lifetime. Some multiple large mtDNA deletions are inherited, as they are caused by mutations in nuclear genes involved in mtDNA replication, nucleotide synthesis or transport [92]. Spontaneous mtDNA deletions appear to accumulate with time, and may increase the likelihood that tissue function will be impaired with time [7].

Above all, mtDNA variants (mutations and deletions) have a broad range of impact, from the ones that are causal of monogenic disorders to those that are a risky allele for complex diseases, in which clinical penetrance also depend on environmental factors. For example, de novo mtDNA monogenic mutations m.3243A > G and m.8344A > G trigger MELAS and MERRF, respectively, and can arise spontaneously, because of replication errors. Another high impact variant is a large 4,977-bp deletion of mtDNA, the most common cause of Kearns–Sayre syndrome. These mutations usually occur de novo within two or three human generations and, for deletions, tend to affect only one generation [104]. In all cases, the specific mutation, the level of heteroplasmy, the threshold effect that determines the level of phenotypic penetrance and which tissues are affected, directly determine the inheritance of the disease and the onset and progression of these disorders.

### 2.2. Mitochondrial Respiratory Chain (MRC) 

The MRC contains five multimeric enzymatic complexes. Complexes I–IV oxidize and transfer their electrons, in the form of hydrogen ions, to the next complex. Once in complex IV, electrons are transferred to molecular oxygen, producing water [7,95]. Complex V generates ATP from adenosine diphosphate (ADP) using the transmembrane electrical potential generated by proton translocation along the MRC [14]. This production of ATP from the reduction of oxygen, known as mitochondrial coupling, is what generates the energy needed for cellular function [110]. In parallel, MRC generates ROS as a subproduct of oxygen metabolism, especially in case of disease. Mutations in mtDNA can trigger the impairment of MRC, affecting energy production and cellular dysfunction [111], observed in some NMDs and mitochondrial diseases [112]. mtDNA mutates more than ten times faster than nuclear DNA, and this high mutation rate in mtDNA is based on the proximity of mtDNA to the inner mitochondrial membrane, where ROS are generated as by-products of MRC chain, coupled with a lack of histones and efficient DNA repair mechanisms. In addition, mtDNA has low number of noncoding regions between genes, leading to random mutations that impact coding sequences of genes responsible of MRC.

In particular, most of the muscular dystrophies (MD) have some MRC complexes with impaired or decreased function (Table 1): complex I in Becker MD (BMD), complexes III and IV in DMD, and complexes I and V in oculopharyngeal (OPMD) and in distal MD (DD). Other NMDs with MRC alterations are myofibrillar myopathies (complexes I and IV) (Table 2), ALS (CIV subunit I) (Table 3), SMA (complexes I and IV) (Table 3), GAN (complexes I and IV) (Table 4), and sIBM (complex IV) [108]. Impaired MRC function promotes mitochondrial dysfunction, eventually leading to progressive weakness and muscle wasting.

To understand the extent of the how MRC dysfunction affects the progression of a NMDs or vice versa, here we present the case of hypothyroid myopathy (endocrine myopathy, Table 2) and ALS (motor neuron diseases, Table 3). 

In hypothyroid myopathy, there is an abnormal activity of the thyroid gland, leading to reduced hormone levels. Thyroid hormones affect skeletal muscle through T3 receptors, located on the mitochondrial membrane of skeletal muscle fibers [113,114]. This MRC-thyroid hormones connection is mediated by several molecular mechanisms that include both direct and indirect effects on mitochondrial structure, function, and biogenesis. One effect in hypothyroid myopathy is a reduced mitochondrial oxidative metabolism, due to decreased hormonal levels and a reduced amount and activity of complex IV (cytochrome c oxidase negative fibers (or COX-)). In addition, T3 receptors, together with the thyroid hormone receptor, increase the expression of some nuclear-encoded respiratory genes, such as cytochrome c1 and b-F1- ATPase subunit genes [38].

Regarding ALS, ALS-associated P525L mutant interacts with ATP synthase complex subunit TP5B, disrupting the formation of ATP synthase complex, thus inhibiting the production of ATP [115]. Both are examples of the complex and intricated pathways governing NMDs in which mitochondria, and herein explained, MRC, play a role.

### 2.3. Mitochondrial Quality Control

#### 2.3.1. Mitochondrial Biogenesis

Mitochondrial biogenesis increases the number of mitochondria, what can lead to a higher energy supply and survival of cells; or if mutated mtDNA expands to higher levels, low energy supply, and cell death. Both outcomes depend on random clonal expansion of mitochondria [103]. Although mitochondrial biogenesis is energy-consuming, it helps to increase energy production and often restores mitochondrial function. Mitochondrial biogenesis can react to many external *stimuli*, such as exercise, hormones, and possibly dietary restriction, in addition to mitochondrial dysfunction [92]. In the case of mitochondrial dysfunction, coordination between mitochondrial synthesis (mitochondrial biogenesis) and degradation (mitophagy) regulates mitochondrial content, quality, and function.

To initiate mitochondrial biogenesis, synthesis of more MRC complexes is required. MRC components are both encoded in nuclear and mtDNA, although their transcripts are not synthesized simultaneously. Cytosolic translation unidirectionally controls mitochondrial translation, both orchestrated by the nuclear genome. The nuclear genes coding for the MRC are rapidly induced under nutrient shift, whereas mitochondrial genes coding for the MRC are induced more slowly. Mitochondrial MRC’s slow kinetics may underlie the lack of environment-responsive mitochondrial transcription factors. Instead of transcription factors, mitochondria contain mRNA-specific translational activators, involved in initiation, elongation, and feedback control of MRC complexes assembly [102]. 

If mitochondrial biogenesis is disrupted, it may result in mitochondrial dysfunction and loss of respiratory capacity [103], observed in patients with mitochondrial diseases. However, excessive mitochondrial biogenesis can also be a sign of disease. In fact, in the muscle of patients with mitochondrial disease, mitochondrial biogenesis generates the so-called ragged-red fibers (RRF), an increase in the number of mitochondria surrounding muscle fibers aimed to compensate mitochondrial dysfunction (Figure 2). RRF typically loss COX activity (partially encoded in the mitochondrial genome) and increase complex II function (succinate dehydrogenase, SDH, entirely encoded in the nuclear genome). Thus, RRF mainly accumulate abnormal proliferated mitochondria [94].

Alterations in some pathways may trigger accumulation of mitochondria in RRF (e.g., changes in mitochondrial protein synthesis). In addition, RRF usually contain a higher amount of mutant mtDNA genomes than non-RRF. Overall, RRF are a tag of defective oxidative phosphorylation in the affected muscle fibers [7]. In fact, RRF are found in many NMDs like limb-girdle MD (LGMD), sIBM, myofibrillar myopathies, myasthenia gravis, ALS, and mitochondrial myopathies (Table 1, Table 2, Table 3 and Table 4). 

#### 2.3.2. Mitochondrial Dynamics

Mitochondria are highly dynamic organelles that continuously fuse, divide, and move, and mitochondrial function is controlled and maintained by these morphologic changes. Mitochondrial fission is mainly mediated by dynamin-related guanosine triphosphatase (GTPase) protein 1 (Drp1); while in mitochondrial fusion, the outer and inner mitochondrial membranes primarily fuse mediated by dynamin-related GTPases mitofusin (Mfn 1 and 2) and optic atrophy 1 (OPA1), respectively [103]. The most direct consequence of mitochondrial division and fusion is the change in size of the mitochondria [105]. Interestingly, these processes promote the exchange of genetic and structural material among participant mitochondria, thus enabling the renewal of damaged components, or its expansion throughout the mitochondrial net (Figure 2). However, mitochondria can also increase in number when there are the specific requirements. 

When mitochondria are damaged, they undergo changes in mitochondrial membrane potential (MMP) causing depolarization. Altered MMP can also impact the mitochondrial fission/fusion machinery and thereby mitochondrial morphology. Mitochondrial fission generates fragmented mitochondria with increased ROS production [106], which may help in the separation of dysfunctional mitochondria from healthy mitochondria and is a prerequisite for mitochondrial degradation (in a process called mitophagy) [107]. The detrimental effect of mitochondrial fission is the release of cytochrome c from mitochondria, which could activate apoptosis signaling through activation of Bax/Bak pore [103]. Apoptotic cell death may be an important mediator of nervous system injury in the mitochondrial encephalomyopathies [7]. Other NMD with increased apoptosis are OPMD, spinal-bulbar muscular atrophy (SBMA), and ion channel diseases. Defects in ion channel diseases may directly reduce MMP and delay repolarization, affecting the modulation of cell survival and increasing cell apoptosis. 

Mitochondrial dynamics is also implicated in the pathogenesis of NMDs like inherited peripheral neuropathy, GAN [76], infantile-onset encephalopathy [116], autosomal dominant optic atrophy [101], and CMT2A [74]. The last three are characterized by mutations involving mitochondrial fusion regulatory genes, OPA1 and MFN2, respectively [117]. Some mutations in mitochondrial dynamics genes can trigger severe effects or even death. An example is a lethal dominant negative allele of DRP1 that affects mitochondrial and peroxisomes fission in newborns and triggers abnormalities in brain development and optic atrophy [118].

Mitochondrial dynamics have also been thoroughly studied in ALS. As mutations in several genes are responsible of ALS, depending on the gene mutated, fusion or fission of mitochondria are enhanced. For example, TDP-43 mutations enhance Mitochondrial fission 1 protein (Fis 1) expression, which increases DRP1 recruitment, leading to higher mitochondrial fission, and a fragmented morphology of mitochondria. In ALS caused by SOD1, increased DRP1 and reduced Mfn1/Opa1 lead to higher level of mitochondrial fission and a fragmented mitochondrial network; while in ALS related to C9ORF72 mutation, higher Mfn1 levels triggered increased mitochondrial fusion and an elongated morphology [119]. Thus, the deregulation of mitochondrial dynamics is frequently associated to NMDs and its proper function protects and counteracts the progression of some NMDs.

#### 2.3.3. Autophagy and Mitophagy

Lysosomes were discovered in 1955 [109], and soon was observed that cytoplasmic components could be located in lysosomes [120]. This was the basis for De Duve (1963) to coin the term autophagy [121]. Autophagy is a lysosomal degradation process characterized by the formation of double-membrane autophagosomes from cytoplasmic material and damaged organelles. Autophagosomes fuse with lysosomes to degrade their content by acidic hydrolases. Autophagic degradation generates amino acids and fatty acids for protein synthesis or oxidation in the MRC to produce ATP for cell survival under starvation conditions. Other functions of autophagy include removal of damaged organelles and protein aggregates, control of cellular biomass and elimination of intracellular pathogens [92]. There are three subtypes of autophagy: macroautophagy, microautophagy, and chaperone-mediated autophagy, divided according to the mechanism used to send intracellular material to be degraded in the lysosomes and the kind of cargo for degradation. 

Briefly, mitophagy—the specific autophagy of mitochondria—initiates with the recognition of damaged mitochondria by the E3-ubiquitin ligase Parkin, which decorates the outer mitochondrial membrane proteins with poly-Ub chains [122]. p62 binds polyubiquitinated proteins and damaged organelles and target them to autophagosomal clearance via its ubiquitin association domain (UBA) and LC3 binding motif (LIR), respectively [123]. Absence of specific mitophagy proteins can block autophagosomal or lysosomal clearance leading to an accumulation of damaged proteins and dysfunctional organelles in autophagosomes [124,125]. 

Mitophagy can occur under several circumstances: metabolic alterations (starvation, particularly), differentiation of some cell types (requiring mitochondrial clearance), and selective targeting and removal of dysfunctional mitochondria [126]. Mitophagy of dysfunctional mitochondria is also called PTEN-induced putative kinase 1 (PINK1)-Parkin-mediated mitochondrial quality-control system [92], since both PTEN and PINK1 are among the main effectors of mitophagy. 

The extent of mitophagy varies in different tissues. For instance, heart, skeletal muscle, nervous system, hepatic and renal tissue have higher mitophagy than thymus and spleen [127]. Defects in mitophagy can lead to accumulation of dysfunctional mitochondria and increased levels of mitochondrial ROS and damage-associated molecular pattern (DAMPs), which are self-molecules that resemble pathogenic ones and can trigger an inflammatory response (Figure 2). Both mitochondrial ROS and DAMPs can initiate an inflammatory response [128]. In addition, impaired mitophagy has been linked with aging, metabolic disorders, cancer, senescence, inflammation, and genomic instability [107]. Regarding NMDs, impaired autophagy has been revealed in congenital MD (CMD), endocrine myopathies, inflammatory myopathies, myofibrillar myopathy and congenital myasthenic syndromes, among others.

In CMD linked to collagen VI deficiency, impaired autophagy is responsible for spontaneous apoptosis, dysfunctional mitochondria and myofibers degeneration. Defective autophagy causes accumulation of abnormal organelles and apoptotic degeneration of muscle fibers. Accumulation of defective mitochondria increases oxidative stress and ROS production, which also contribute to increase apoptosis. The cause of defective autophagy in CMD is a reduced protein amount of beclin-1 and Bnip3, what leads to a defective activation of the autophagy process [15]. 

As mitophagy protects from the accumulation of defective mitochondria and ROS overproduction, this process is frequently impaired in NMDs affecting both nerve and muscle function.

### 2.4. Mitochondrial ROS and ER Stress

ROS comprises superoxide anion (O_2_^•−^), hydrogen peroxide (H_2_O_2_), hydroxyl radical (OH^•^), peroxyl radical (ROO^•^), and nitric oxide (NO^•^). Although mitochondrial oxidative metabolism is a major source of ROS in many cell types, outside mitochondria there are other enzymes associated with ROS production, like NADPH oxidase, neural nitric oxide synthase, and monoamine oxidase [129]. A balance between oxidative species and antioxidant defense mechanisms is required for cell homeostasis [130]. Briefly, the antioxidant response is mediated by enzymes responsible for metabolizing or neutralizing ROS, such as catalase, glutathione peroxidase, or superoxide dismutase (SODs) [131]. 

The high proportion of ROS generation in MRC is due to the transfer of electrons in the inner mitochondrial membrane until they encounter oxygen, when electrons are converted in water. Under hypoxia or pathologic conditions this process is not completed, resulting in an increase of superoxide anions [131]. For example, mutation in SOD2, an antioxidant enzyme encoded in the mitochondrial genome, triggers the development of CMT, a NMD [132]. Mutations in SOD1 gene are found in ALS. SOD1 encodes the Cu-Zn superoxide dismutase, one of three proteins involved in the conversion of free superoxide radicals to molecular oxygen and hydrogen peroxide. Mutations in the SOD1 gene are found in 10–20% of familial ALS cases and 1–5% of sporadic ALS cases globally [133]; so far, more than 170 mutations of the SOD1 gene are known in ALS [119]. 

Other NMDs in which high ROS levels are not compensated and cells hold excessive oxidative stress are BMD, DMD, FSHD, ALS, SBMA, mitochondrial myopathies, and FA. 

When overproduction of ROS cannot be compensated by antioxidant defenses or mitophagy, ER receives ROS signals through MAMs, which may eventually activate cell stress responses to compensate and overcome this adverse situation [134].

ER is the main organelle in charge of protein biosynthesis and folding and one of the main cell sensors to stress insults. Alterations of the ER redox status can negatively affect protein folding and result in ER stress. ER stress occurs when the rate at which new protein entering the ER exceeds its folding capacity, which can lead to the activation of three transmembrane proteins: inositol-requiring enzyme 1α (IRE1α), PKR-like ER kinase (PERK), and the activation transcription factor 6 (ATF6), which altogether trigger a UPR [134]. The UPR is comprised of a series of transcriptional, translational, and post-translational processes that can lower the rate of protein synthesis and increase the protein folding capacity of the ER machinery. This results in an increase of misfolded protein elimination, and escalating the size of the ER compartment thereby reducing the ER stress [98]. Usually, transient ER stress can be overcome by UPR, but if the damaging stimulus persists, inflammatory response genes are activated [131].

ER-mitochondria crosstalk is involved in several pathways including cell proliferation, apoptosis, autophagy, lipid metabolism, Ca^2+^ signaling, UPR, inflammation, and bioenergetics. Alterations in this ER-mitochondria crosstalk are associated with multiple diseases, like motor neuron diseases, myotonic dystrophy, OPMD, endocrine myopathies, myofibrillar myopathies, SBMA, CMT, and mitochondrial myopathies. In ALS, over expression of both wild-type and mutated TDP-43 leads to a decrease in ER-mitochondria physical and functional coupling, affecting Ca^2+^ regulation. Ca^2+^ dysregulation is considered to be a primary cause of motor neuron death in ALS [135]. Ca^2+^ is responsible for the link between metabolic impairment and MAMs interaction. This is seen in cancer cells, which are able to remodel their intracellular Ca^2+^ signaling, enhancing their survival and proliferation. Modulation of ER-mitochondrial Ca^2+^ crosstalk favors resistance to apoptosis [136].

In addition to triggering ER stress, if ROS damage overwhelms these repair mechanisms, MMP may decrease, thus potentially leading to the increase in the permeability of mitochondrial membranes. Higher permeability in mitochondrial membrane can activate the mitochondrial permeability transition pore (MPTP) and thus, the release of DAMPs to the cytosol (mtDNA), ceramides, formaldehyde) [137,138]. MPTP and ROS can also promote the activation of mitochondrial fission and mitophagy to eliminate damaged mitochondria (Figure 2). In mitochondrial fission, MAMs signal the ER tubules to encircle mitochondria and tag the sites for mitochondrial division [139]. Accordingly to mitochondrial implication in NMDs, dysregulation of MPTP opening and defective autophagy have been associated with muscular dystrophies, like collagen VI myopathies (CMD), BMD, DMD, LGMD, and DD [14,15] and, remarkably, most of NMDs show some level of deregulation in most of these pathways. 

### 2.5. Mitochondrially Induced Inflammatory Response

Inflammatory responses recognize pathogen-associated molecular patterns (PAMPs), derived from infection, through a variety of receptors. However, because of the bacterial origin of mitochondria, their proteins are structurally similar to those in bacteria and enable their recognition by the same receptors of the immune system, reinforcing the notion of mitochondria as hubs of immunity [140]. Thus, mitochondrial DAMPs can also trigger an inflammatory response [128]. DAMPs include proteins and peptides, such as N-formyl peptides and TFAM, as well as lipids, and metabolites such as cardiolipin, succinate and ATP, and mtDNA [139].

#### 2.5.1. Implication of mtDNA in Inflammation

In particular, mtDNA is recognized by multiple innate immune receptors: cytosolic cyclic GMP-AMP synthase (cGAS), endosomal Toll-like receptor 9 (TLR9), and NOD, LRR, and Pyrin domain-containing protein 3 (NLRP3) [140]. Here we briefly remark the main effects of these innate immune responses.

Cytosolic mtDNA enhances cytosolic antiviral signaling and expression of interferon-stimulated genes (IRF3/IRF7), which results in the activation of cGAS DNA sensor and STING-IRF3-dependent signaling [141]. These pathways activate transcription factors NF- kB and IRF3 through the kinases IKK and TBK1, respectively. MtDNA stress triggered by TFAM deficiency has been reported to release mtDNA to the cytosol [101]. 

TLR9 is a nucleic acid sensor that binds to unmethylated CpG DNA, like mtDNA. In basal conditions, TLR9 is located in the ER, but when it is stimulated, TLR9 translocates to the membrane of endosomes or lysosomes, where it binds to ligands and triggers cell inflammation [140].

TLR9 interaction with mtDNA occurs in the endolysosomal compartment and activates the myeloid differentiation primary response protein 88 (MyD88), which induces a number of kinases and transcriptional factors, namely mitogen-activated protein kinases (MAPK) [101]. MAPK leads to nuclear factor-kB (NF-κB) and IRF7 activation to enhance pro-inflammatory and interferon type I (IFN-1) responses, respectively [89]. NF-kB signaling increases the expression of other pro-inflammatory cytokines such as tumor necrosis factor-a (TNF-a), interleukin (IL)-6, IL-1b [128]. Furthermore, TLR9 senses incomplete digestion in lysosomes of damaged mitochondria, because of impaired mitophagy, without the mtDNA release to cytosol.

The activation of the inflammasome NLRP3 involves two sequential signals. The inflammasome priming and assembly signal is induced by NF-κB, which acts downstream of TLRs and other immune receptors [142]. NF-kB activates the expression of inflammasome components and inactive forms of the cytokines, including pro-IL-1b. When active, NLRP3 inflammasome activates caspase-1, which processes pro-IL-1β and pro-IL-18 [100,143]. DAMPs directly activate NLRP3 inflammasome as well. Mitochondria are suggested to regulate the activity of the NLRP3 inflammasome complex, by: activating mitophagy (reduces inflammation by clearing mitochondrial-bound NLRP3 complexes) and releasing mitochondrial ROS (amplifies inflammasome immunogenic signal) [140].

In normal conditions, NLRP3 protein is located on the ER. Under oxidative stress and in response to inflammation, NLRP3 can be translocated to MAMs. MAMs transmit danger signals through physical interaction, promoting coordinated responses to oxidative stress, such as triggering mitophagy to remove damaged mitochondria [131]. 

Altogether, mitochondria-induced inflammation usually is initiated by mitochondrial ROS, which can cause the activation of MAPKs by inhibiting MAPK phosphatase. Activated MAPK may aid in the production of IL-6 and TNF. Mitochondrial ROS accumulation can also activate NLRP3 inflammasome, which promotes the maturation of IL-1β and IL-18. mtDNA accumulation in the cytosol can interact with TLR-9 (in lysosomes) to induce inflammatory responses (e.g., in autosomal dominant optic atrophy [101]. This pathologic feature has been described in sIBM and in LMNA-related NMDs (CMD, Emery-Dreifuss MD, LGMD, CMT, etc.)) [144]. Further, cytosolic mtDNA can activate cGAS-STING to induce IFN-1 or the NLRP3 inflammasome which can induce the maturation/secretion of pro-inflammatory cytokines [101]. Thus, mtDNA can activate major innate immune responses by acting as a DAMP from the mitochondria. Overall, mtDNA is released from mitochondria to the cytosol, where it is recognized by immune receptors, which trigger a signaling cascade that leads to the production of cytokines or transcription of inflammatory genes in the nucleus (Figure 2) [143]. This close association between mitochondria and inflammation triggered from affected cells, explains why NMDs are usually associated with inflammatory effects. 

#### 2.5.2. Implication of Mitochondrial ROS in Inflammation

Regarding mitochondrial ROS, they are sensed by mitochondrial antiviral signaling protein (MAVS), key in viral RNA infections. MAVS can activate pathways that regulate NF-kB and IRF3/IRF7 to induce gene expression [145]. It also can interact with the outer mitochondrial membrane, where mitochondrial ROS can trigger MAVS oligomerization, promoting IFN-1 production. In addition, MAVS can associate with NLRP3 and triggers its oligomerization, leading to caspase-1 activation [146]. 

NLRP3 is activated by mitochondrial ROS but also by cardiolipin, an inner mitochondrial membrane lipid. Cardiolipin translocates to the outer mitochondrial membrane after mitochondrial membrane depolarization, where it can recruit NLRP3. Cardiolipin-NLRP3 interaction suggests the role of mitochondria as a hub for the activation of innate immunity [143].

Similar to MAVS, NLRP3 also responds to mitochondrial ROS and can cause mitochondrial damage that promotes the generation of additional ROS. NLRP3 drives the production of IL-1β, IL-18, and pyroptosis. mtDNA can also activate NLRP3, and is also sensed by TLR9, which leads to immune and inflammatory gene expression. Mitochondria are therefore critical for signaling by three major innate immune pathways: RIG-I/MAVS, NLRP3, and TLR9 [143]. All these pathways rely on mitochondria-inflammatory response and, consequently, are altered in numerous NMDs.

#### 2.5.3. Feedback Regulation of the Mitochondrial Inflammatory Response 

NF-kB has a role as a pro-inflammatory and anti-inflammatory transcription factor in mitochondrial-induced inflammation. NF-kB anti-inflammatory functions prevent premature and excessive NLRP3-inflammasome activation, avoiding chronic inflammatory disease and inflammation caused by cell and tissue stress [147]. Self-limiting inflammation is also important for maintaining homeostasis, tissue repair, and regeneration, which restores integrity of epithelial barriers and thereby attenuates PAMP and DAMP availability [100]. 

Autophagy is the mechanism underlying NF-kB-mediated inhibition of NLRP3 inflammasome. After inducing the expression of pro-inflammatory cytokines and NLRP3, NF-kB is able to switch the initial pro-inflammatory to an anti-inflammatory state to avoid excessive inflammation. Thus, NF-kB induces the expression of p62, which eliminates NLRP3 by mitophagy [138]. Therefore, the “NF-kB-p62/SQSTM1-mitophagy” pathway provides an essential regulatory loop through which NF-kB orchestrates a reparative inflammatory response and prevents excessive collateral damage [100]. 

Dysregulation of the NLRP3-inflammasome is frequently associated with diverse inflammatory, metabolic, and malignant diseases, including gouty arthritis, Alzheimer’s disease, obesity, type II diabetes, and colorectal cancer [148]. Other NMDs with increased inflammation are ALS (with aberrant activation of NF-kB) [17], muscular dystrophies (like BMD and DMD), and inflammatory myopathies. Therefore, proper control of NLRP3-inflammasome activity and overall inflammatory cascade are critical for preventing disease development. 

The aging process itself is linked to elevated low-grade inflammation or para-inflammation, characterized by constitutive production of low amounts of IL-1b, TNF, and IL-6. The NLRP3-inflammasome activation and insufficient mitophagy link accumulation of damaged mitochondria to the para-inflammatory state. Normal, healthy aging may be compromised and greatly enhanced by accumulation of somatically mutated mitochondria, which are more likely to release mitochondrial ROS and fragmented mtDNA that act as direct NLRP3-inflammasome activators [100]. This explains why most of NMDs arise hand by hand to aging processes.

## 3. Treatment Strategies in NMDs

As stated before, mitochondria play a key role in the early process of nerve and muscle degeneration and affects the progression of NMDs. Although current understanding of mitochondrial dysfunction in NMD is still evolving, there are some potential therapeutic strategies that tackle mitochondrial deregulated pathways.

The number of NMDs for which treatments are available is limited. This scarcity of therapeutic options lays on the rarity and diversity of NMDs, the complexity of their genetic inheritance, the abundance and distribution of muscle tissue, and low treatment efficacy [14]. The rarity of NMDs limits the available patients for recruitment for clinical trials, which highly depends on the quality and reliability of data obtained in cell and animal experiments [149]. 

Despite there is no cure for most NMDs, some can be effectively managed and treated. Some common interventions include drug therapy, surgery, or patient support groups, to improve physical and psychological wellbeing of NMD patients [6]. Innovative breakthroughs are approaching owing to antisense oligonucleotides (ASO) and gene replacement therapies, currently under development. Here we summarize the main points of different treatment strategies, most of them posing supportive pharmacologic options aimed not to cure but to slow down disease progression.

Drug therapy in NMDs comprises a wide range of treatments. Immunosuppressive drugs are widely used to treat certain muscle, nerve, and NMJ diseases. Anticonvulsants and antidepressants might aid in treating the pain of neuropathy [6]. More specifically, senolytic drugs (e.g., resveratrol) help to remove senescent cells to reduce the overall pro-inflammatory profile, although inflammation differs among tissues and makes it difficult to only target senescent cells [82]. Interestingly, resveratrol has been associated to promote mitochondrial health and renewal through different mechanisms [150].

Other treatment options are being developed to specifically target mitochondria. For example metformin, used as an anticancer drug, increases the activation of AMPK, favoring autophagy of damaged mitochondria (e.g., in mitochondrial diseases and DMD); caloric restriction, which reduces ROS production and thus, DNA damage (in collagen VI myopathies); and mitohormesis, based in inducing mild mitochondrial stress during life to induce an antioxidant response (e.g., by vitamin E). Some dietary supplements boost mitochondrial capacity like niacin (vitamin B3) in mitochondrial myopathy; epicatechin in BMD, or triheptanoin, CoQ10 and tocotrienols in ALS [151].

Moreover, exercise has shown to induce autophagy to overcome sarcopenia (e.g., in metabolic myopathies) [152]. In young and old mice, exercise training has even promoted biogenesis of mitochondria and mitochondrial autophagy to reduce the expression of proinflammatory cytokines by up to 49% [82]. Unfortunately, exercise is not always possible in patients affected by NMDs.

Other mitochondrial pathways that could be tackled therapeutically are dysregulation of MPTP opening, mitochondrial fission, and ER stress. MPTP modifiers (e.g., CsA, cyclophilin D inhibitors) might potentially aid in many NMDs [14]. Targeting mitochondrial fission is based on its direct relation to apoptosis, thus reducing fission might reduce cell death. Treating ER stress could be relevant in many NMD and other proteinopathies [30]. Currently, these therapeutic approaches are under development.

In addition to drug therapy, some NMDs benefit from surgery. Depending on the NMD, patients may receive neurological, thoracic, orthopedic, or other surgeries. Physical, occupational, or rehabilitation therapies may help before or after these interventions [6].

Recently, other approaches more related to personalized medicine are being developed or even in the market, like ASO and gene therapy. ASOs are short, synthetic single-stranded nucleic acids that bind targeted RNAs sequences to promote their degradation or modify splicing by impeding the binding of RNA splicing factors [153]. This approach has been developed in SMA and DMD and approved by FDA. In SMA, an ASO that switches the splicing to include an exon in the pre-mRNA of SMN2 gene has become the first treatment for SMA approved by FDA, commercialized by Biogen (MA, USA) under the name Nusinersen [67]. 

In DMD, an exon skipping ASO called Eteplirsen is also approved and commercially available. In this case, exon-skipping of damaged dystrophin in DMD generates a shortened but still functional version of dystrophin [154]. Other NMDs like LGMD could benefit from ASO exon skipping, by restoring the disrupted reading frame of the SGCG gene [5,27]. 

Apart from ASOs, gene replacement therapies are under development. In SMA, there is a gene replacement therapy available for patients younger than 2 years old. In this case, SMN1 gene cDNA is packaged in an adeno-associated virus 9 vector (AAV9), which are non-pathogenic, delivered intravenously and can cross the blood brain barrier to transduce neurons [155]. Once inside neurons, SMN1 cDNA constitutively expresses, with long-term expression in differentiated cells after a single dose, but not in mitotically active cells. This drug has shown efficacy in infantile SMA patients and is commercialized by Avexis under the name Zolgensma [67,156]. Other gene therapies using CRISPR/Cas-mediated approaches in NMD have shown efficacy in models of DMD, CMD, LGMD, DM1, and FSHD [157], becoming promising therapeutic approaches for further transference into clinical settings. 

Altogether, currently there are nearly 200 potential therapies under development for NMD (in preclinical and clinical stages), mainly targeting ALS, SMA, and DMD [158]. Around 43% are small molecules (targeting receptor modulation, epigenetic reprogramming, redox metabolism, etc.), 14% gene therapy and 9% antisense oligonucleotides (ASOs) [159]. This shows that personalized medicine is increasingly offering new treatments for rare diseases, like NMDs. Indeed, the number of molecules in clinical trials for NMD has sharply increased in the last 5 years [158]. This proves NMD therapies are becoming a highly dynamic and potentially achieving field [8].

Notably, treatments not only focus in improving physical health of NMD patients, but also patients and families receive education and counselling to check their wellbeing. This may include getting involved in support groups or genetic counsellors (e.g., physical therapists, nutritionists). Patient’s associations are highly interested in creating support groups and analyze the quality of life of NMD patients and carers. One ongoing project, promoted by Share4Rare and released in the beginning of 2020, consists in a study called “Understanding how neuromuscular diseases impact learning and working opportunities for patients and carers.” They claim that patients with rare NMDs are a scattered community, often dispersed, and poorly represented. Even with the increased interest in developing research and potential therapeutic options in clinical trials in the last 10 years, little is known about the impact on quality of life of NMDs. With this study, they focus on understanding the psychosocial impact of NMD on patients, families, and caregivers [160].

## 4. Discussion

Despite we must acknowledge the impossibility of summarizing all previous knowledge in the current review, the present revision aimed to outline the state-of-the-art in the main etiologies of NMDs related to mitochondrial metabolism. The role of oxidative stress, mitochondrial biogenesis, autophagy, and inflammation related to mitochondria have been thoroughly described and linked to the affected NMDs (Figure 2). Overall, mitochondrial defects play a key role in the early process of neuronal and muscle degeneration and further affects the progression of NMDs. 

NMDs can appear as a result of damage in nerve, muscle, or both tissues. In fact, the classification of NMDs include seven groups, two of them with primary muscle affection (MD and myopathies other than dystrophies), three of them with primary nerve dysfunction (NMJ diseases, peripheral nerve diseases and motor neuron diseases) and last two with both tissues affected (mitochondrial diseases and ion channel diseases). More attention should be focused on identifying prognostic and diagnostic biomarkers that target a specific pathway to accurately diagnose every NMD and control its progression.

Despite the number of treatments for NMD is still limited, some experimental treatments are currently under development to tackle mitochondrial deregulated pathways in NMDs. Current studies and clinical trials in many disabling NMDs aim to improve the wellbeing of patients and their families, but deep understanding of mitochondrial dysfunction in NMD will be essential to validate and establish further therapeutic targets.

The preclinical and clinical research advances in NMDs in the last years have been astounding. Better cell and animal models have helped in the characterization of many NMDs. Furthermore, the broad diversity and rarity of these diseases have revealed the importance of carefully studying every NMD history and progression. This more accurate approach is translating to more personalized treatments, as seen with new ASOs and gene therapies recently released to the market, and will eventually lead to a higher success rate in clinical trials for these disabling diseases.

## 5. Conclusions

NMDs have a direct or indirect influence of mitochondria in their etiology. Mitochondrial genetic defects directly affect nerves and muscles in NMDs; but when there is an alternative cause of disease, unproper mitochondrial function can limit energetic supply of compensatory mechanisms, thus indirectly conditioning the progression of disease.Defects in oxidative metabolism can lead to accumulation of ROS, potentially affecting the ER and triggering inflammation, which eventually may lead to cell death and tissue damage.To avoid cell damage and excessive oxidative stress, mitochondrial quality control processes closely monitor changes in mitochondrial metabolism. In case of trouble, different responses arise by the increase in the number of mitochondria (mitochondrial biogenesis), by changing their morphology and size (mitochondrial dynamics) or recycling them (mitophagy). Most of these processes are altered in NMDs, highlighting the relevance of mitochondria in the development and progression of these disorders.Understanding nerves, muscles, and mitochondrial defects in NMDs is essential to improve diagnosis and treatments for these major incapacitating diseases.Despite growing efforts to clarify the etiology of NMD, the complexity of reality is still far beyond current knowledge and information provided in the present review, challenging new researchers to develop novel approaches.

## Figures and Tables

**Figure 1 antioxidants-09-00964-f001:**
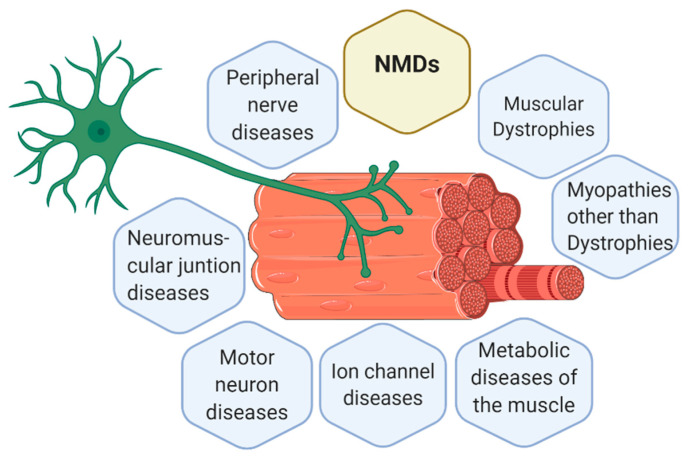
Representation of a motor unit according to the classification of the main neuromuscular diseases (NMDs). Muscle cells and neurons interact in the neuromuscular junction (NMJ) where muscle cells receive the instructions from the lower motor neurons to perform the actions required. Thus, body movement is actioned by muscle fibers but controlled by the nervous system. Consequently, alterations in muscle fibers or nerves can both trigger a NMD. Figure created with BioRender.com.

**Figure 2 antioxidants-09-00964-f002:**
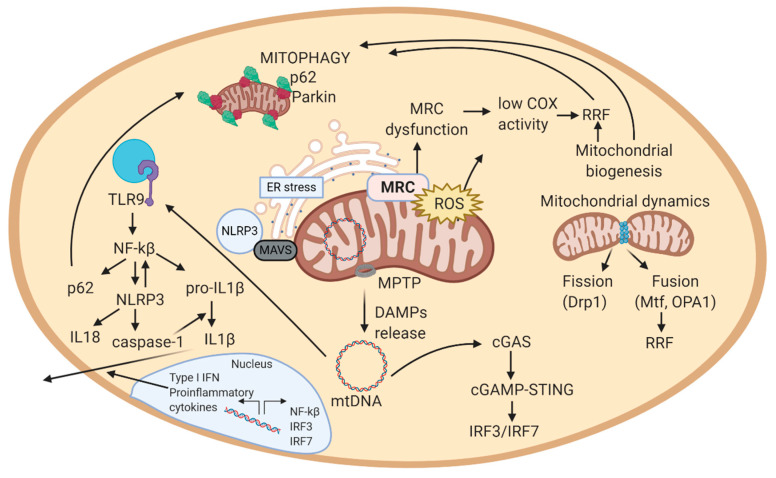
Schematic view of the mechanistic model of mitochondrial-related pathways in a cell. 1. MRC generates ATP in the inner mitochondrial membrane. In this process, also ROS are produced. 2. In case of excessive ROS levels, mitochondrial membrane potential decreases, allowing MPTP to open and release DAMPs (mtDNA, cardiolipin, ceramides) to the cytosol. These DAMPs can trigger an inflammatory response. In this scheme, the inflammatory cascade represented is the one activated by mtDNA. mtDNA activates cGAS in cytosol, that eventually induces the expression of interferon-related genes (IRF3/IRF7). In parallel, mtDNA activates TLR9, which activates NF-kB, and thus, NLRP3 inflammasome, expression of pro-inflammatory cytokines, and among them, IL-1 response. 3. Increased ROS levels can result from MRC dysfunction. To compensate dysfunctional mitochondria, mitochondrial biogenesis increases the number of mitochondria in a cell. Excessive mitochondria around muscle cells form RRF. 4. Excessive ROS, RRF, inflammation, and ER stress damage mitochondria, which alterations are sensed by autophagy and mitophagy. Autophagy and mitophagy remove damaged mitochondria, and consequently stop the inflammatory response, supporting repair mechanisms in the cell. If autophagy and mitophagy are altered, mitochondria cannot be recycled, leading to apoptosis. Abbreviations: MRC: mitochondrial respiratory chain; ROS: reactive oxygen species; MPTP: mitochondrial permeability transition pore; DAMPs: damage-associated molecular patterns; mtDNA: mitochondrial DNA; cGAS: cyclic GMP-AMP synthase; TLR9: toll-like receptor 9; NF-kB: nuclear factor kappa-light-chain-enhancer of activated B cells; NLRP3: NOD-like receptor protein 3; RRF: ragged-red fibers; ER: endoplasmic reticulum; MAVS: mitochondrial antiviral signaling protein. Figure created with BioRender.com.

**Table 1 antioxidants-09-00964-t001:** Examples of studies reporting altered mitochondrial functions in muscular dystrophies.

NMD Group	Name of Main Diseases	Description of Principal Disease Features	Description of Main Mitochondrial Alteration	Most Affected Mitochondrial Pathways	Main Evidence Level/Disease Model	Relevant References	Examples of Mutations in Genes
**Muscular Dystro-phies (MD)**	**Becker MD (BMD)**	Atrophy of the skeletal, cardiac, and pulmonary muscles	Reduced mitochondrial mass, complex I activity and ATP levels, and increased Ca^2+^ levels	Increased MPTP opening, ROS levels, and inflammation	in vitro, cells, animal (*C. elegans*, *mdx* mice), patient (CT)	[11,12,13]	DMD
	**Congenital MD (CMD)**	Muscle weakness and possible joint deformities with slow progression and shortened life span	Most common CMD are collagen VI myopathies, with visible mitochondrial dysfunction	Increased MPTP opening and defective autophagy	in vitro, cells, animal (Col6a1 -/- mice), patient (CT)	[2,14,15]	LMNA, DPM3, DAG1, TRAPPC11
	**Duchenne MD (DMD)**	General muscle weakness and wasting due to lack of dystrophin protein. Shortened lifespan. Rarely affects women (milder symptoms and better prognosis)	Massive aggregates of mitochondria, lower activities of MRC complexes III and IV and increased Ca^2+^.	Increased MPTP opening, ROS levels and inflammation	in vitro, cells, animal (*C. elegans*, *mdx* mice), patient (CT)	[8,11,12,14,16,17,18,19,20,21]	DMD
	**Emery-Dreifuss MD (EDMD)**	Muscular weakness and atrophy of shoulder, upper arm, and shin muscles, with early joint contractures and cardiomyopathy	Reduced expression of MRC complex genes and upregulation of mitochondrial disassembly genes, altered mitochondrial location and morphology	Decreased MRC and altered mitochondrial biogenesis	in vitro and animal models (*C. elegans*, mice)	[2,22,23,24]	LMNA, EMD, FHL1
	**Faciosca-pulohume-ral MD (FSHD)**	Muscle weakness that affects mainly facial, shoulder, and arm muscles	Reduced antioxidative response molecules (low levels of zinc, selenium, and vitamin C)	Higher ROS and mitochondrial dysfunction	cells and patient (CT)	[14,17,25,26]	TRPV4, DUX4, SMCHD1
	**Limb-girdle MD (LGMD)**	Weakness and wasting of the muscles in hips and shoulders	Morphologic mitochondrial abnormalities, including RRF and decreased COX	MPTP dysregulation and mitochondrial dysfunction	in vitro, cells, animal (*521ΔT* mice), patient (CT)	[2,5,14,27]	SGCG, LMNA, DYSF, TCAP, TRIM32, TNPO3 *
	**Myotonic dystrophy (MD)**	Muscle loss and weakness due to inability to relax them. It affects facial muscles first, but also feet, hands, and neck	Altered mitochondrial proteins (decreased EF-Tu, hsp60, GRP75, dienoyl CoA isomerase) and disruption of ubiquitin-proteasome systems in MD2 (affects proximal muscles)	ER stress and mitochondrial dysfunction	cells and patient (CT)	[14,28,29]	CNBP (ZNF9), DMPK
	**Oculopha-ryngeal MD (OPMD)**	Weakness of eye, face, and throat muscles leading to drooping eyelids and problems with swallowing	PABPN1 protein aggregates, reduced complex I and V mitochondrial proteins, altered UPR and apoptosis	ER stress, mitochondrial dysfunction and apoptosis	in vitro, cells, animal, patient (CT)	[24,30,31,32]	PABPN1
	**Distal MD (DD) (or Distal myopathy)**	Weakness and wasting of muscles of the hands, forearms, and lower legs with slow progression. Many DD diseases	Decreased mitochondrial membrane potential, increased mitochondrial oxygen consumption and Ca^2+^ and deficiencies in MRC complexes I and IV	MPTP opening and ATP depletion	in vitro, cells, patient (CT)	[14,33,34]	FLNC, TTN, DYSF

Family of muscular dystrophies (MD) and their respective disease features, mitochondrial alterations, affected mitochondrial pathways, evidence level of altered mechanisms, mutations in causal genes and relevant references. Examples of genes with mutations causal/related to a MD taken from Muscle gene table (accessed July 2020) [35]. * More LGMD mutations: HNRNPDL, CAPN3, COL6A1, SGCA, SGCB, SGCG, SGCD, POMT1, POMT2, POMGNT1, POMGNT2, FKTN, ANO5, COL6A2, COL6A3, DAG1, PLEC, TRAPPC11, GMPPB, ISPD, POGLUT1, DNAJB6, FKRP, TTN, LAMA2, BVES. Abbreviations: MPTP: mitochondrial permeability transition pore; ROS: reactive oxygen species; CT: clinical trials; RRF: ragged-red fibers; COX: cytochrome c oxidase, MRC complex IV; ER: endoplasmic reticulum; UPR: unfolded protein response.

**Table 2 antioxidants-09-00964-t002:** Examples of studies reporting altered mitochondrial functions in myopathies other than dystrophies.

NMD Group	Name of Main Diseases	Description of Principal Disease Features	Description of Main Mitochondrial Alteration	Most Affected Mitochondrial Pathways	Main Evidence Level/Disease Model	Relevant References	Examples of Mutations in Genes
**Myopa-thies**	**Congenital myopathies**	Inherited diseases that affect the tone and contraction of skeletal muscles causing general muscle weakness	The most frequent congenital myopathies are core myopathies, with reduced MRC activity and near-total depletion of mitochondria	reduced MRC and mitochondrial depletion	in vitro, cells, animal, patient (CT)	[14,29,36,37]	RYR1, DNM2, MTM1, TNPO3, COL6A1/2/3, COL12A1, PYROXD1, MSTO1
	**Endocrine myopathies**	Weakness and atrophy (shrinking) of the muscles around the shoulders and hips, muscle stiffness, cramps, and slowed reflexes caused by abnormal activity of the thyroid gland. Two types: hypothyroid (reduced hormone levels) and hyperthyroid myopathies (excess in hormones)	Hypothyroid myopathy: decreased TFAM, reduced mitochondrial DNA copy number and mitochondrial alterations (COX- fibers). Hyperthyroid: moderate increase in mitochondrial size and protein aggregates	mitochondrial structure, ER stress (proteotoxicity), autophagy failure	cells	[38,39,40]	
	**Inflammatory myopathies**	Chronic muscle inflammation accompanied by prolonged muscle fatigue and weakness. sIBM is the inflammatory myopathy with more mitochondrial alteration	Abnormal mitochondria in sIBM (RRFs and COX- fibers). In sIBM and PM, mtDNA deletions in muscle and altered autophagy	Altered mitochondrial structure, mtDNA deletions and autophagy	in vitro, cells, animal, patient (CT)	[7,14,41,42]	VCP, HNRNPA1, GNE
	**Metabolic myopathies**	Group of disorders caused each by a different genetic defect that impairs the body’s metabolism causing muscle weakness, exercise intolerance, muscle pain or cramps	Altered MRC: substrates are not properly processed or cannot enter mitochondria affecting energy production of the cell. mtDNA or nDNA mutations	reduced MRC and ATP depletion	in vitro, cells, patient (CT)	[7,28,43]	GAA, AGL, GBE1, PYGM, PFKM, PHKA1
	**Myofibrillar myopathies (MFM)**	Characterized by muscle weakness, cardiomyopathy, myalgia, loss of sensation and weakness in the limbs (peripheral neuropathy), and respiratory failure	Mitochondrial abnormalities with RRF and enlarged mitochondria. Protein aggregates affecting distribution and function of mitochondria. Deficiencies in MRC complex I and IV	MRC dysfunction, autophagy and ER stress, reduced mitochondrial biogenesis	in vitro, cells, patient	[14,44,45,46]	MYOT, DES, PYROXD1, CRYAB, LDB3, FLNC, BAG3, TTN
	**Scapulopero-neal myopathy**	Rare genetic disorder characterized by weakness and wasting of specific muscles: shoulder blade area (scapula) and the smaller of the two leg muscle groups below the knee (peroneal)	mtDNA mutations reported in a few cases	mtDNA	cells	[47,48,49]	VCP, FHL1

Family of myopathies other than dystrophies and their respective disease features, mitochondrial alterations, affected mitochondrial pathways, evidence level of altered mechanisms, mutations in causal genes and relevant references. Examples of genes with mutations causal/related to a myopathy taken from muscle gene table (accessed July 2020) [35]. Abbreviations: MRC: mitochondrial respiratory chain; CT: clinical trials; TFAM: mitochondrial transcription factor A; COX-: cytochrome c oxidase negative fibers; ER: endoplasmic reticulum; sIBM: sporadic Inclusion Body Myositis; RRF: ragged-red fibers; PM: polymyositis; mtDNA: mitochondrial DNA; nDNA: nuclear DNA.

**Table 3 antioxidants-09-00964-t003:** Examples of studies reporting altered mitochondrial functions in Neuromuscular Junction Diseases (NMJ) and Motor Neuron Diseases.

NMD Group	Name of Main Diseases	Description of Principal Disease Features	Description of Main Mitochondrial Alteration	Most Affected Mitochondrial Pathways	Main Evidence Level/Disease Model	Relevant References	Examples of Mutations in Genes
	**Congenital myasthenic syndromes (CMS)**	Weakness and fatigue resulting from problems at NMJ. Different types of CMS, according to the part of the NMJ affected: presynaptic (the nerve cell), postsynaptic (the muscle cell) or synaptic (the space in between)	Gene defect in mitochondrial citrate carrier SLC25A1 underlie deficits in NMJ transmission. SLC25A1 is involved in many biological processes (e.g., glycolysis, autophagy)	NMJ signaling	in vitro, cells, animal (zebrafish, mice), patient (CT)	[50,51,52,53]	CHRNA1, PLEC, CHRNB1, CHRND, CHRNE, COLQ, CHAT, SYT2, AGRN, SLC5A7, SYT2
**Neuro-muscular junction diseases (NMJ)**	**Lambert-Eaton myasthenic syndrome (LEMS)**	Autoimmune disease that attacks the calcium channels in the NMJ and interferes with the ability of nerve cells to send acetylcholine to muscle cells, affecting muscle contraction and causing muscle weakness	Altered calcium channels	NMJ signaling	cells, patient (CT)	[50,54,55,56]	SYT2
	**Myasthenia gravis (MG)**	Chronic autoimmune disorder in which antibodies destroy neuromuscular connections. It affects voluntary muscles of the body, especially the eyes, mouth, throat, and limbs	Mitochondrial morphological alterations and RRF	mitochondrial morphology, neuromuscular connections	cells, animal (rat, mice), patient (CT)	[55,57,58]	CHAT
	**Amyotrophic lateral sclerosis (ALS)**	Fatal disease with degeneration of nerve cells in the spinal cord and brain. It affects voluntary control of arms and legs and eventually leads to trouble breathing	Accumulation of mitochondrial in proximal axons, mitochondrial injury by ROS excess, COX I mtDNA mutation and RRF	Increased ROS and altered mitochondrial structure	in vitro, cells, animal (*SOD1G93A* mice), patient (CT)	[7,14,17,18,45,59,60,61,62,63,64,65]	SOD1, ALS2, SPG111, HNRNPA1, SQSTM1*
**Motor neuron diseases**	**Spinal-bulbar muscular atrophy (SBMA)**	Genetic disorder in which loss of lower motor neurons affect voluntary muscle movement, specifically facial, swallowing muscles and limbs. Only affects men	Reduced MMP. Increased expression of apoptotic proteins that activate mitochondrial caspase pathway. AR unfolding and oligomerization induces toxicity	MPTP opening, increased ROS and apoptosis. Lower mitochondrial mass and ER stress	in vitro, cells, animal, patient (CT)	[66,67,68,69,70]	AR
	**Spinal muscular atrophy (SMA)**	Genetic disease affecting the central and peripheral nervous system, and voluntary muscle movement, mainly shoulders, hips, thighs, and upper back	Decreased enzyme activities involving MRC complexes I-IV causing mitochondrial dysfunction	mitochondrial dysfunction and altered MRC	in vitro, cells, animal, patient (CT)	[8,14,16,17,18,67,69,70,71]	SMN1, IGHMBP2, SIGMAR1, PLEKHG5, DNAJB2, VRK1, TRPV4

Family of neuromuscular junction diseases (NMJ) and motor neuron diseases and their respective disease features, mitochondrial alterations, affected mitochondrial pathways, evidence level of altered mechanisms, mutations in causal genes and relevant references. Example of genes with mutations causal/related to a NMJ diseases taken from muscle gene table (accessed July 2020) [35]. * More ALS mutations: ubiquilin 2, C9orf72, SIGMAR1, CHCHD10, C9orf72, VCP, TDP-43, SETX, FUS, VAPB. Abbreviations: NMJ: neuromuscular junction; CT: clinical trials; RRF: ragged-red fibers; ROS: reactive oxygen species; COX I: cytochrome c oxidase subunit I; mtDNA: mitochondrial DNA; MMP: mitochondrial membrane potential; AR: androgen receptor; MPTP: mitochondrial permeability transition pore; ER: endoplasmic reticulum.

**Table 4 antioxidants-09-00964-t004:** Examples of studies reporting altered mitochondrial functions in peripheral nerve diseases and mitochondrial diseases.

NMD Group	Name of Main Diseases	Description of Principal Disease Features	Description of Main Mitochondrial Alteration	Most Affected Mitochondrial Pathways	Main Evidence Level/Disease Model	Relevant References	Examples of Mutations in Genes
**Peripheral nerve diseases**	**Charcot-Marie-Tooth disease (CMT)**	Inherited disorder that affects nerves outside of your brain and spinal cord, nerves that supply feet, legs, hands, and arms. Two subtypes: CMT1 (demyelinating), CMT2 (axonal)	Altered mitochondrial dynamics and axonal transport of mitochondria, causing axonal degeneration. Mutations in mitochondrial fusion regulatory genes	mitochondrial dynamics and ER stress	in vitro, cells, animal, patient (CT)	[14,17,26,45,67,72,73,74]	CMT1: PMP22, MPZ. CMT2: MFN2, LMNA, VCP, DNM2, IGHMBP2, DNAJB2, AARS, DYNC1H1, SOD2
	**Giant axonal neuropathy (GAN)**	Inherited condition characterized by abnormally large and dysfunctional axons. First, limbs have problems with walking, followed by difficulties coordinating movements (ataxia), and require wheelchair assistance	Abnormal and enlarged mitochondria in Schwann cells, deficiencies of complexes I and IV, several mtDNA point mutations and multiple mtDNA deletions.	altered mitochondrial size and mitochondrial dynamics, reduced MRC function	in vitro, cells, animal, patient (CT)	[7,75,76]	GAN
**Mitochon-drial diseases**	**Mitochon-drial myopathies**	Genetic defects that affect mitochondria can cause muscular and neurological problems (e.g., muscle weakness, exercise intolerance, trouble with balance and coordination). Many mitochondrial myopathies are known	Mutations in mtDNA or nDNA affecting proteins involved in MRC, mitochondrial morphology (RRF) and protein aggregation (UPR)	Altered energy production and redox signaling, mitochondrial morphology and ER stress	in vitro, cells, animal (*ANT-* mice), patient (CT)	[9,77,78,79]	MRPS25, TIMM22, NDUFAF1, COX6A2, SLC25A42
	**Friedreich’s ataxia (FA)**	Muscle weakness and ataxia, loss of balance and coordination due to reduced synthesis of the mitochondrial protein frataxin. It mostly affects the spinal cord, peripheral nerves and cerebellum	Altered synthesis of frataxin affects mitochondrial iron metabolism and homeostasis and antioxidant protection	oxidative stress, mitochondrial dysfunction	in vitro, cells, patient (CT)	[7,80,81]	FXN

Family of peripheral nerve diseases and mitochondrial diseases and their respective disease features, mitochondrial alterations, affected mitochondrial pathways, evidence level of altered mechanisms, mutations in causal genes and relevant references. Example of genes with mutations causal/related to a peripheral nerve disease or mitochondrial disease taken from muscle gene table (accessed July 2020) [35]. Abbreviations: ER: endoplasmic reticulum; CT: clinical trials; mtDNA: mitochondrial DNA; MRC: mitochondrial respiratory chain; nDNA: nuclear DNA; RRF: ragged-red fibers; UPR: unfolded protein response.

**Table 5 antioxidants-09-00964-t005:** Examples of studies reporting altered mitochondrial functions in ion channel diseases.

NMD Group	Name of Main Diseases	Description of Principal Disease Features	Description of Main Mitochondrial Alteration	Most Affected Mitochondrial Pathways	Main Evidence Level/Disease Model	Relevant References	Examples of Mutations in Genes
**Ion channel diseases (or Channe-lopathies)**	**Andersen-Tawil syndrome**	Altered potassium channel gene affect the heartbeat and the ability of muscles to stay ready to contract. Paralysis may occur	Mitochondrial channelopathies affect the K^+^, Ca^2+^, VDAC and MPTP channels. Reduced channel activity rate results in reduced MMP and delayed repolarization, causing mitochondrial dysfunction. Intracellular Ca^2+^ homeostasis, mitochondrial bioenergetic metabolism, and modulation of cell survival and death are also affected	reduced MMP, mitochondrial dysfunction, apoptosis	in vitro, cells, animal (mice, rat, *C. elegans*, patient (CT)	[82,83,84,85]	KCNJ2
	**Hyperkalemic periodic paralysis**	Genetic alterations in sodium channels results in temporary muscle weakness, and eventually, temporary paralysis					SCN4A, T704M, M1592V
	**Hypokalemic periodic paralysis**	Genetic defects in calcium or sodium channel cause a loss of muscle excitability when serum potassium is low					SCN4A, CACNA1S, ATP1A2, KCNE3
	**Myotonia congenita**	Disease caused by mutations in the gene encoding a chloride channel necessary for stopping muscle contraction. Delayed muscle relaxation triggers muscle stiffness					CLCN1 (CLC-1), SCN4A
	**Paramyotonia congenita**	Mutations in the muscle sodium channel gene prolong the channel’s opening, higher muscle excitation triggering episodes of muscle stiffness and weakness, mostly in the face, neck and upper extremities					SCN4A
	**Potassium-aggravated myotonia (or Sodium Channel myotonias)**	Sustained muscle tensing causes muscle stiffness that worsens after exercise and may be aggravated by eating potassium-rich foods					SCN4A

Family of ion channel diseases and their respective disease features, mitochondrial alterations, affected mitochondrial pathways, evidence level of altered mechanisms, mutations in causal genes and relevant references. Example of genes with mutations causal/related to an ion channel disease taken from muscle gene table (accessed July 2020) [35]. Abbreviations: VDAC: voltage-dependent anion channels; MPTP: mitochondrial permeability transition pore; MMP: mitochondrial membrane potential.

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
