# Peer review of "The Impact of Mitochondrial Deficiencies in Neuromuscular Diseases"

_antioxidants, 2020, doi:10.3390/antiox9100964_

Round 1
Reviewer 1 Report
In the manuscript by Cantó-Santos et al the authors provide a review on the metabolic alterations underlying neuromuscular diseases (NMDs) mainly focusing on the mitochondria.
The manuscript addresses interesting contributing factors to the pathogenesis of NMDs and seeks to provide the relation amongst them. The review is well structured in terms of the topics addressed and the authors made an effort to include the reviewers suggestions having improved the quality of the manuscript which I feel now that can be accepted for publication should the authors include some minor suggestions that follow:
line 51 - Mitochondrial respiratory chain is now written but the abbreviation is still missing
line 52 - use "indicate" or "suggest" instead of "proof"
line 98 - line 101 It was nice to include this paragraph. It greatly benefited the text
line 110 - Attention this paragraph is again an oversimplified version. Therefore not very accurate. Please rephrase it
line 377 - Should be "Briefly, mitophagy - the specific autophagy of mitochondria- initiates with the...."
Author Response
In the manuscript by Cantó-Santos et al the authors provide a review on the metabolic alterations underlying neuromuscular diseases (NMDs) mainly focusing on the mitochondria.
The manuscript addresses interesting contributing factors to the pathogenesis of NMDs and seeks to provide the relation amongst them. The review is well structured in terms of the topics addressed and the authors made an effort to include the reviewers suggestions having improved the quality of the manuscript which I feel now that can be accepted for publication should the authors include some minor suggestions that follow:
We kindly appreciate the effort of revising again the manuscript to improve its quality. We thank the reviewer for highlighting the key points and we corrected the minor suggestions that still needed to be addressed. The lines are numbered according to the clean version of the manuscript.
line 51 - Mitochondrial respiratory chain is now written but the abbreviation is still missing
MRC abbreviation eventually included (line 50).
line 52 - use "indicate" or "suggest" instead of "proof"
We substituted proof by indicate (line 50).
line 98 - line 101 It was nice to include this paragraph. It greatly benefited the text
Thank you for the feedback and for the careful revision.
line 110 - Attention this paragraph is again an oversimplified version. Therefore not very accurate. Please rephrase it
We rephrased it to improve its accuracy and readability (line 109).
line 377 - Should be "Briefly, mitophagy - the specific autophagy of mitochondria- initiates with the...."
We included this information in the sentence (line 378).
Reviewer 2 Report
The updated version of the manuscript has corrected all the weak points of the previous version. I have some minor comments to it:
Line 45-46: My previous comment was not understood. I meant to write the following sentence: “Blood tests can determine abnormal levels of various common metabolites and antigens in the blood in certain NMDs”.
Line 51: First citation, please spell out: Mitochondrial Respiratory Chain (MRC)
Line 70: First citation, please spell out: Neuromuscular junction (NMJ) diseases
Line 104-105: please rephrase
Line 108-109: please rephrase
Line 178: “reducing the health”? Or do you mean “reducing both health and economic burdens”?
Line 264: a mutant of which gene/protein? Please, specify.
Lines 339-340: rephrase. Development of what?
Line 379: … target them…
Line 410: Please, change the X for the actual formulation: peroxyl radical (ROO•)
Lines 447-448: “affecting Ca2+ dysregulation”? Do you mean “affecting Ca2+ regulation”?
Line 523: Do you mean “Interferon type 1” (IFN-1)?
Author Response
The updated version of the manuscript has corrected all the weak points of the previous version. I have some minor comments to it:
We strongly appreciate the effort of revising again the manuscript. In this version, we have corrected the minor comments that still needed to be addressed. The lines are numbered according to the clean version of the manuscript.
Line 45-46: My previous comment was not understood. I meant to write the following sentence: “Blood tests can determine abnormal levels of various common metabolites and antigens in the blood in certain NMDs”.
We corrected the sentence. We are sorry for the previous misunderstanding (line 44).
Line 51: First citation, please spell out: Mitochondrial Respiratory Chain (MRC)
MRC abbreviation has been included (line 50).
Line 70: First citation, please spell out: Neuromuscular junction (NMJ) diseases
We changed the word order accordingly (line 69).
Line 104-105: please rephrase
We rephrased it to improve its accuracy and readability (line 103).
Line 108-109: please rephrase
We rephrased it to improve its accuracy and readability (line 107).
Line 178: “reducing the health”? Or do you mean “reducing both health and economic burdens”?
We meant “reducing both health and economic burdens”. Thank you for pointing out this issue (line 178).
Line 264: a mutant of which gene/protein? Please, specify.
We included two genes as examples (line 264).
Lines 339-340: rephrase. Development of what?
We changed “development” to “function” and “progression” to specify the sentence and improve its comprehension (line 340-341).
Line 379: … target them…
We removed the termination of the third person of the present simple (the final “s”) from the verb (“target”) (line 381). Thanks for such a precise checking.
Line 410: Please, change the X for the actual formulation: peroxyl radical (ROO•)
We updated the formulation of the peroxyl radical (line 412).
Lines 447-448: “affecting Ca2+ dysregulation”? Do you mean “affecting Ca2+ regulation”?
Yes, we corrected it to Ca2+ regulation (line 450). Thank you for pointing out.
Line 523: Do you mean “Interferon type 1” (IFN-1)?
This sentence has been revised according to reviewer suggestion (line 523).
This manuscript is a resubmission of an earlier submission. The following is a list of the peer review reports and author responses from that submission.
Round 1
Reviewer 1 Report
The review submitted by Canto-Santos and col is focus on the role of mitochondria in the different neuromuscular diseases. Mitochondria are not only important for aerobic energy production in the cell, but they also are key in other processes such as apoptosis or the regulation of ROS production. Being the crossroad of many key functions, when defective, critically contribute to those diseases.
The review is well structured, and very complete. However, there are some points that I would like to indicate:
- the Figure 1 is very nice, but it is bringing new information, when compared to the point 1.1.1 from the main text. If the authors want to keep the figure, I would suggest them to include indications to the aberrations that are there written.
- In the whole text, the motorneurons from the spinal cord should be called "lower motor neurons", to clearly differentiate them from the "upper motor neurons" located in the cerebral cortex.
- line 83: authors state that nervous system and skeletal muscle have low cell turnover. In the muscle we find satellite cells for renewal, and the peripheral nervous system has regenerative capabilities. This sentence should be rephrase, if they want to indicate that neurons and sarcomers do not proliferate.
- I would suggest to change "Ca+2" by "Ca2+"
- The complete section 2 is not required, better remove it.
- line 246-247: please, revise this sentence to make it more precise. Do the authors mean "the nuclear genes coding for the MRC..."
- lines 283-289: where exactly TLR9 interacts with the mtDNA? In the lysosome or in the cytosol? This is confusing. In the same sense, the sentence 410-412 is also confusing.
- line 523: please correct to SMN2. And also, in line 530, which SMN gene? SMN1 or SMN2?
- line 562: are mitochondrial defects only responsible for muscle degeneration? Or are they also involve in neuronal degeneration in NMDs?
Minor:
- line 43: "are focused"
- line 45-46: "various common metabolites and ..."
- line 70: "transmission of signals between nerves and muscles"
- line 81: section "2.1"
- line 111: "are essential"
- line 235: "mtDNA expands"
- line 251: "is disrupted"
- NF-kb or NF-kB? Please, unify
Authors Response
The review submitted by Canto-Santos and col is focus on the role of mitochondria in the
different neuromuscular diseases. Mitochondria are not only important for aerobic energy
production in the cell, but they also are key in other processes such as apoptosis or the
regulation of ROS production. Being the crossroad of many key functions, when defective,
critically contribute to those diseases.
The review is well structured, and very complete.
The reviewer has kindly summarized the key points of the review and has highlightened its
structure and completeness. We sincerely appreciate the effort of the reviewer in carefully
revising the manuscript.
However, there are some points that I would like to indicate:
- the Figure 1 is very nice, but it is bringing new information, when compared to the point 1.1.1
from the main text. If the authors want to keep the figure, I would suggest them to include
indications to the aberrations that are there written.
We agree with the reviewer’s suggestions. As Figure 1 contains information explained in sections
1.1.1 and 1.2, we decided to mention the figure also in section 1.2 (line 109). This way we reflex
all information contained in Figure 1 within the text of the review.
- In the whole text, the motorneurons from the spinal cord should be called "lower motor
neurons", to clearly differentiate them from the "upper motor neurons" located in the cerebral
cortex.
We corrected the nomenclature of “lower motor neurons” in the manuscript and we appreciate
this suggestion for improving the specificity and accuracy of the text.
- line 83: authors state that nervous system and skeletal muscle have low cell turnover. In the
muscle we find satellite cells for renewal, and the peripheral nervous system has regenerative
capabilities. This sentence should be rephrase, if they want to indicate that neurons and
sarcomers do not proliferate.
We specified the difference between specialized neurons and muscle cells from their reservoir of
nerve and muscle stem cells (lines 93-97). We did previously not consider stem cells in this
paragraph and we appreciate the point made by the reviewer.
- I would suggest to change "Ca+2" by "Ca2+"
We acknowledge this correction in the terminology of ions.
- The complete section 2 is not required, better remove it.
We remove section 2 but we included a paragraph in the end of section 1.3. to point out the aim
of the review before starting the main body of the manuscript.
- line 246-247: please, revise this sentence to make it more precise. Do the authors mean "the
nuclear genes coding for the MRC..."
We rephrased the sentence to make it more accurate, as stated by the reviewer (line 284). Thanks
for improving the understanding of the review.
- lines 283-289: where exactly TLR9 interacts with the mtDNA? In the lysosome or in the cytosol?
This is confusing. In the same sense, the sentence 410-412 is also confusing.
As stated by the reviewer, TLR9 interacts with the mtDNA in the lysosome. It is therefore specified
in the text (lines 487). The same applies to sentences in lines 510-512.
- line 523: please correct to SMN2. And also, in line 530, which SMN gene? SMN1 or SMN2?
In line 613), we corrected the misspelling of SMN2. In line 620 and 623 we pointed that the gene
involved is SMN1, not previously mentioned.
- line 562: are mitochondrial defects only responsible for muscle degeneration? Or are they also
involve in neuronal degeneration in NMDs?
Mitochondrial defects are also involved in neuronal degeneration (line 653). We appreciate this
suggestion that was missing in the sentence.
Minor:
- line 43: "are focused"
Corrected (line 43)
- line 45-46: "various common metabolites and ..."
“Common” included in the phrase (line 45)
- line 70: "transmission of signals between nerves and muscles"
We changed the word order accordingly (line 71)
- line 81: section "2.1"
We corrected it from “1.1” to “1.2”. We consider it part of the Introduction (section “1.2”) (line
90).
- line 111: "are essential"
Corrected (line 130)
- line 235: "mtDNA expands"
Corrected (line 273)
- line 251: "is disrupted"
Corrected (line 290)
- NF-kb or NF-kB? Please, unify
Unified as NF-kB
Reviewer 2 Report
This reviewer general comments are the following:
In the manuscript by Cantó-Santos et al the authors aim to provide a review on the metabolic alterations underlying neuromuscular diseases (NMDs) mainly focusing on the mitochondria. Although the review also aims to discuss in more detail oxidative stress, autophagy and inflammation processes that are closely related to the mitochondria I believe the title should be more focused on mitochondria.
The manuscript addresses interesting contributing factors to the pathogenesis of NMDs and seeks to provide the relation amongst them. However, there are some shortcomings that require the authors’ attention:
- The review is well structured in terms of the topics addressed, however to my opinion all the topics are too general and do not focus enough on NMDs. Most of them do not bring any new views/perspectives to the subjects addressed and merely describe well known concepts (such as mitochondrial dynamics or autophagy) only briefly referring to the processes in the context of NMDs in one or two sentences and with very few references to the original research works. The authors claim in the discussion that "The role of oxidative stress, mitochondrial biogenesis, autophagy and inflammation related to mitochondria have been thoroughly described and linked to the affected NMDs" but the link of all the described processes to particular NMDs is frequently not clear throughout the text. The exception is to the topic on treatment strategies that is fairly well documented and covering several aspects regarding the available therapies.
- In several situations throughout the text ideas are generalized/ extrapolated, missing a bit accuracy e.g. p.4 line 150 "If mitophagy is impaired, excessive ROS affects the ER, through mitochondria-ER membrane contact sites, called Mitochondria Associated Membranes (MAMs), triggering ER stress [26]." Although the concept is not incorrect it oversimplifies the pathways/mechanisms and passes the idea that this is the only mechanism by which these processes occur.
Minor comments and notes:
p2 line 51 - MRC was not previously abbreviated, except in the abstract
p2 line 87 - it is not accurate to say that motor neurons are located in the peripheral nervous system, since they are also present in the brain (midbrain).
p5 line 216 - MRC generates ROS not exactly due to abnormal oxygen metabolism, but as a subproduct of normal oxygen metabolism
p6 line 244 - "transcripts levels are not synthesized simultaneously" should be "transcripts are not synthesized simultaneously"
p6 line 252 - do not use "contrarily"
p7 line 304 - should be mitophagy instead of autophagy, as the mechanism described does not apply to autophagy in general
p7 line 308 - this sentence is too general
p8 line 313 - use "several" instead of "these" since there are also other circumstances (aside the ones that are described) under which mitophagy occurs
p8 line 328 - attention H2O2 is hydrogen peroxide and not "oxygen peroxide" as it is stated
p8 line 346 - either use "its folding capacity" or "the folding capacity within the ER"
p8 line 348 - it should not be used "undergo" but instead use "drive" or "trigger", since IRE1a, PERK and ATF6 do not undergo UPR, they are the primary sensors and effectors of the UPR
p9 line 382 - this sentence is missing a reference
p9 line 399 - MAM has been previously abbreviated (line 150)
Note 1: The tables are very helpful in summarizing the state of the art in NMDs, and therefore I consider them a plus in this paper.
Note 2: To improve readability the text should be checked for language/grammar.
Note 3: In some parts the text appear to have very different "modes of writing". I understand that this might happen when several authors write a paper but the authors should try to make the text more "homogeneous"
Note 4: Figure 2 starts to be referred on page 6 but only appears on page 10. Would be nice to have it before.
Note 5: The paper would benefit from "stronger" conclusions
Authors Response
In the manuscript by Cantó-Santos et al the authors aim to provide a review on the metabolic
alterations underlying neuromuscular diseases (NMDs) mainly focusing on the mitochondria.
Although the review also aims to discuss in more detail oxidative stress, autophagy and
inflammation processes that are closely related to the mitochondria I believe the title should be
more focused on mitochondria.
The reviewer has kindly summarized the aims of the review. We sincerely appreciate the effort
made by the reviewer to perform a careful revision of the manuscript to improve its
understanding and specificity. In line with this, the reviewer suggests a title more focused on
mitochondria. We suggest this alternative: “The impact of mitochondrial deficiencies in
Neuromuscular Diseases”.
The manuscript addresses interesting contributing factors to the pathogenesis of NMDs and
seeks to provide the relation amongst them.
We value the reviewer’s insights and acknowledgements of our review contribution.
However, there are some shortcomings that require the authors’ attention:
- The review is well structured in terms of the topics addressed, however to my opinion
all the topics are too general and do not focus enough on NMDs. Most of them do not
bring any new views/perspectives to the subjects addressed and merely describe well
known concepts (such as mitochondrial dynamics or autophagy) only briefly referring to
the processes in the context of NMDs in one or two sentences and with very few
references to the original research works. The authors claim in the discussion that "The
role of oxidative stress, mitochondrial biogenesis, autophagy and inflammation related
to mitochondria have been thoroughly described and linked to the affected NMDs" but
the link of all the described processes to particular NMDs is frequently not clear
throughout the text. The exception is to the topic on treatment strategies that is fairly
well documented and covering several aspects regarding the available therapies.
We acknowledge that the relationship between some processes in the context of NMDs could
have been explained with more detail. In the revised version, we provide a more accurate
description of the links between a specific pathway and NMDs with more references to the
original research works (for example in lines 257-264; lines 234-242, etc.). Thanks again for this
upgrade.
- In several situations throughout the text ideas are generalized/ extrapolated, missing a
bit accuracy e.g. p.4 line 150 "If mitophagy is impaired, excessive ROS affects the ER,
through mitochondria-ER membrane contact sites, called Mitochondria Associated
Membranes (MAMs), triggering ER stress [26]." Although the concept is not incorrect, it
oversimplifies the pathways/mechanisms and passes the idea that this is the only
mechanism by which these processes occur.
We revised the manuscript and rephrased some sentences that were too general and wrongly
simplified the complexity of reality. It was the consequence of summarizing all the information
in a limited space. Unfortunately, we must admit that given the broad range of pathways and
processes involved in mitochondria and NMDs, each one with their particularities and
idiosyncrasy, probably we still not fully cover every situation in detail. This has been
acknowledged at the end of the manuscript (see lines 687-689), as follows: ‘Despite growing
efforts to clarify the etiology of NMD, the complexity of reality is still far beyond current
knowledge and information provided in the present review, challenging new researchers to
develop novel approaches’.
Some examples of sentences that we have rephrased or adapted (with modal verbs) to reduce
generalizations are:
- Lines 430-432 (rephrased): When overproduction of ROS can not be compensated by
antioxidant defenses or mitophagy, ER receives ROS signals through MAMs, which may
eventually activate cell stress responses to compensate and overcome this adverse situation [69].
- Lines 507-509-(modal verbs included): Altogether, mitochondria-induced inflammation usually
is initiated by mitochondrial ROS, which can cause the activation of MAPKs by inhibiting MAPK
phosphatase. Activated MAPK may aid in the production of IL-6 and TNF.
Minor comments and notes:
p2 line 51 - MRC was not previously abbreviated, except in the abstract
Corrected (line 51)
p2 line 87 - it is not accurate to say that motor neurons are located in the peripheral nervous
system, since they are also present in the brain (midbrain).
The reviewer is absolutely right. We acknowledge the correction that has been amended in the
new version of the manuscript to improve the specificity and accuracy of the text. To differentiate
the motor neurons from the spinal cord to the ones from the midbrain we called them “lower
motor neurons”.
p5 line 216 - MRC generates ROS not exactly due to abnormal oxygen metabolism, but as a
subproduct of normal oxygen metabolism
Corrected (line 239)
p6 line 244 - "transcripts levels are not synthesized simultaneously" should be "transcripts are
not synthesized simultaneously"
Corrected (line 282)
p6 line 252 - do not use "contrarily"
It was changed to “however” (line 291)
p7 line 304 - should be mitophagy instead of autophagy, as the mechanism described does not
apply to autophagy in general
Changed to mitophagy (line 379)
p7 line 308 - this sentence is too general
It has been rephrased (line 383)
p8 line 313 - use "several" instead of "these" since there are also other circumstances (aside the
ones that are described) under which mitophagy occurs
Right. Changed to “several” (line 385)
p8 line 328 - attention H2O2 is hydrogen peroxide and not "oxygen peroxide" as it is stated
Absolutely. Corrected (line 411)
p8 line 346 - either use "its folding capacity" or "the folding capacity within the ER"
We used “its folding capacity” (lines 435-436)
p8 line 348 - it should not be used "undergo" but instead use "drive" or "trigger", since IRE1a,
PERK and ATF6 do not undergo UPR, they are the primary sensors and effectors of the UPR
Thanks. We used “trigger” (line 438)
p9 line 382 - this sentence is missing a reference
Reference added (line 483)
p9 line 399 - MAM has been previously abbreviated (line 150)
Thanks. Corrected
Note 1: The tables are very helpful in summarizing the state of the art in NMDs, and therefore I
consider them a plus in this paper.
We kindly appreciate the feedback.
Note 2: To improve readability the text should be checked for language/grammar.
We carefully revised the manuscript again to improve its readability
Note 3: In some parts the text appear to have very different "modes of writing". I understand
that this might happen when several authors write a paper but the authors should try to make
the text more "homogeneous"
We revised the manuscript again and tried to make it more uniform.
Note 4: Figure 2 starts to be referred on page 6 but only appears on page 10. Would be nice to
have it before.
We agree and moved Figure 2 to page 8 (after section 2.3.2), close to where it is cited by the first
time.
Note 5: The paper would benefit from "stronger" conclusions
We revised the conclusions to make them stronger, thanks again for helping us to improve the quality of the review.